# Storage of Transfusion Platelet Concentrates Is Associated with Complement Activation and Reduced Ability of Platelets to Respond to Protease-Activated Receptor-1 and Thromboxane A2 Receptor

**DOI:** 10.3390/ijms25021091

**Published:** 2024-01-16

**Authors:** Linnea I. Andersson, Dick J. Sjöström, Huy Quang Quach, Kim Hägerström, Lisa Hurler, Erika Kajdácsi, László Cervenak, Zoltán Prohászka, Erik J. M. Toonen, Camilla Mohlin, Tom Eirik Mollnes, Per Sandgren, Ivar Tjernberg, Per H. Nilsson

**Affiliations:** 1Department of Chemistry and Biomedicine, Linnaeus University, 391 82 Kalmar, Sweden; linnea.andersson@lnu.se (L.I.A.); dick.sjostrom@lnu.se (D.J.S.); camilla.mohlin@lnu.se (C.M.); 2Mayo Clinic Vaccine Research Group, Mayo Clinic, Rochester, MN 55905, USA; huy.q.quach@gmail.com; 3Department of Clinical Chemistry and Transfusion Medicine, Region Kalmar County, 391 85 Kalmar, Sweden; kim.hagerstrom@regionkalmar.se (K.H.); ivar.tjernberg@liu.se (I.T.); 4Department of Internal Medicine and Haematology, Semmelweis University, 1088 Budapest, Hungary; lisa.hurler@web.de (L.H.); kajdacsi.erika@med.semmelweis-univ.hu (E.K.); cervenak.laszlo@gmail.com (L.C.); prohaszka.zoltan@med.semmelweis-univ.hu (Z.P.); 5R&D Department, Hycult Biotechnology, 5405 Uden, The Netherlands; e.toonen@hycultbiotech.com; 6Department of Immunology, Oslo University Hospital, University of Oslo, 0372 Oslo, Norway; t.e.mollnes@medisin.uio.no; 7Research Laboratory, Nordland Hospital, 8005 Bodo, Norway; 8Center for Hematology and Regenerative Medicine (HERM), Karolinska Institutet, 171 77 Huddinge, Sweden; per.sandgren@regionstockholm.se; 9Department of Biomedical and Clinical Sciences, Division of Inflammation and Infection, Linköping University, 581 83 Linköping, Sweden; 10Linnaeus Centre for Biomaterials Chemistry, Linnaeus University, 391 82 Kalmar, Sweden

**Keywords:** platelet storage, platelet storage lesion, complement activation, platelet function, hemostasis

## Abstract

Platelet activation and the complement system are mutually dependent. Here, we investigated the effects of storage time on complement activation and platelet function in routinely produced platelet concentrates. The platelet concentrates (n = 10) were stored at 22 °C for seven days and assessed daily for complement and platelet activation markers. Additionally, platelet function was analyzed in terms of their responsiveness to protease-activated receptor-1 (PAR-1) and thromboxane A2 receptor (TXA_2_R) activation and their capacity to adhere to collagen. Complement activation increased over the storage period for all analyzed markers, including the C1rs/C1-INH complex (fold change (FC) = 1.9; *p* < 0.001), MASP-1/C1-INH complex (FC = 2.0; *p* < 0.001), C4c (FC = 1.8, *p* < 0.001), C3bc (FC = 4.0; *p* < 0.01), and soluble C5b-9 (FC = 1.7, *p* < 0.001). Furthermore, the levels of soluble platelet activation markers increased in the concentrates over the seven-day period, including neutrophil-activating peptide-2 (FC = 2.5; *p* < 0.0001), transforming growth factor beta 1 (FC = 1.9; *p* < 0.001) and platelet factor 4 (FC = 2.1; *p* < 0.0001). The ability of platelets to respond to activation, as measured by surface expression of CD62P and CD63, decreased by 19% and 24% (*p* < 0.05) for PAR-1 and 69–72% (*p* < 0.05) for TXA_2_R activation, respectively, on Day 7 compared to Day 1. The extent of platelet binding to collagen was not significantly impaired during storage. In conclusion, we demonstrated that complement activation increased during the storage of platelets, and this correlated with increased platelet activation and a reduced ability of the platelets to respond to, primarily, TXA_2_R activation.

## 1. Introduction

Platelets are recognized for their role in hemostasis and inflammation. Still, they are transfused primarily for their hemostatic function to patients suffering from thrombocytopenia because of illness, traumatic bleeding, or medication [1]. Depending on the diagnosis, the Association for the Advancement of Blood & Biotherapies recommends starting prophylactic transfusions at a platelet count of 10–50 × 10^9^/L [2]. Platelets circulate in the bloodstream and respond to vascular damage by adhering to the exposed subendothelial matrix. Platelets can bind directly to exposed collagen via glycoprotein (GP) VI and GP Ia/IIa [3] or to von Willebrand Factor via GP Ib-IX-V [4]. During this process, platelets undergo activation and respond by releasing intracellularly stored alpha and dense granules, expressing a distinct set of activation markers, including CD62P and CD63. Adenosine diphosphate (ADP) and thromboxane A_2_ (TXA_2_), released from activated platelets, and thrombin generated by the cleaving of prothrombin on the surface of activated platelets by the prothrombinase complex, further activate platelets in the vicinity [5]. Altogether, this ultimately leads to blood loss prevention by forming a platelet plug and stimulation of coagulation, which includes the build-up of a fibrin network and modulation of the innate immune response.

The isolation of blood components for storage ex vivo implies an adjusted environment in terms of temperature, surface contact, and extracellular fluid composition. Blood banks in Sweden commonly store platelet concentrates for up to seven days [6], even though variations exist between centers and national guidelines worldwide [7]. In the Random Donor Platelets (RDPs) from the Reveos^®^ Automated Blood Processing System from Terumo BCT (Larne, UK), the platelet concentrates are stored at 22 °C for up to seven days in Platelet Additive Solution-E (PAS-E) containing a mixture of NaCl, Na_3_ citrate, NaH_2_PO_4_, Na_2_HPO_4_, Na acetate, KCl, and MgCl_2_. The platelet concentrate typically consists of 1/3 (*v*/*v*) plasma originating from all included donors [8], which exposes it to a risk of accumulation of various inflammatory mediators and cell debris in the plasma fraction upon storage, including complement activation products.

The complement system is a protein-based cascade system containing plasma and membrane-bound components [9]. Soluble complement pattern recognition molecules, including C1q which activates the classical pathway, or mannose-binding lectin (MBL), ficolin-1, -2, and -3, which activate the lectin pathway, can be activated in contact with artificial surfaces [9,10], including in platelet concentrates during storage. Activation can progress through complement components C4, C2, and C3 cleavage. After C3 cleavage, the generated C3b can initiate the alternative complement pathway including factor D cleaving factor B, which amplifies C3 cleavage [11]. All three pathways converge downstream of C3 into the terminal pathway with C5 cleavage and the formation of C5a and the terminal C5b-9 complement complex. Throughout activation of the complement system, effector molecules are generated, of which, soluble C3a and C5a and the surface bound C3b/iC3b are potent fragments for cell surface receptor activation.

Complement activation and platelet activation are mutually dependent. Complement activation products C3a and C5a have been shown to activate platelets [12,13], even though the impact of complement anaphylatoxins on platelets and the expression of the corresponding receptors on platelets have not been fully resolved [14]. The impact of C3b on platelet aggregation was recently shown [15], and the insertion of the terminal C5b-9 membrane attack complex through the platelet membrane mediates platelet cytolysis with the release of DAMPs [14]. This may fuel a vicious cycle where platelets can activate complement in the fluid phase via the release of intracellular stored chondroitin sulfate to activate the classical pathway [16]. Strong platelet activation can also lead to complement activation on the platelet surface [17,18], even though this event is tightly regulated by the binding of complement regulators [19]. Activated platelets can also mediate non-proteolytic activation of C3 into C3(H_2_O) [20], which, in turn, can recruit properdin to the platelet surface [21].

Complement activation in stored platelet concentrates is linked to platelet activation and cytolysis, with the liberation of damage-associated molecular patterns (DAMPs) that are associated with adverse transfusion reactions in patients [22]. The complement system is at risk of activation during ex vivo plasma isolation and storage if not appropriately inhibited by efficient Ca^2+^ (classical and lectin pathway) [23] and Mg^2+^ chelation (all pathways) [24,25] and cold storage [26]. Here, we comprehensively characterized plasma complement activation, including the novel assays for classical and lectin pathway activation, cytokine levels, platelet activation markers, and a platelet functional analysis in routine transfusion platelet concentrates stored for up to seven days.

## 2. Results

### 2.1. Minor Platelet Consumption during the Storage Period

Ten Random Donor Platelets (RDPs) were analyzed in the present study. Three concentrates were prepared from whole blood and the platelets were processed into Interim Platelet Units (IPUs) on the same day; three were stored overnight before processing; and four were a mix between IPUs processed from overnight-stored and fresh whole blood. When they left the blood bank, the ten RDPs had a mean platelet number of 859 × 10^9^/L with a standard deviation of 111 × 10^9^/L. None of the platelet bags had bacterial contamination. The leukocyte count ranged from 0.19 to 0.73 × 10^6^/L, which were below the acceptable limit of 1.0 × 10^6^ per 300 mL RDP. The platelet number was recorded every day during the storage period. There was a small consumption of platelets during the storage period (Figure 1). At Day 7, the number of platelets had dropped significantly (*p* < 0.05) by 2.4%.

### 2.2. Increased Complement Activation during Storage

A comprehensive analysis of complement activation markers was performed in the T_0_ samples (n = 10) from Day 1 to Day 7. Both the classical pathway-specific marker C1s/C1-INH and the lectin-specific marker MASP-1/C1-INH were significantly increased after 24 h of storage (*p* < 0.001 and *p* < 0.05, respectively) (Figure 2A,B). Compared to Day 1, the levels for C1s/C1-INH increased 1.9-fold (*p* < 0.0001) and MASP-1/C1-INH by 2.0-fold (*p* < 0.0001) after the seven-day storage. Consistently, C4c, an activation marker of both classical and lectin pathway activation, rose by 1.8-fold (*p* < 0.0001) by Day 7 (Figure 2C). C3bc, reflecting C3 cleavage activity irrespectively of the activation pathway, increased by 4.0-fold (*p* < 0.01) during storage (Figure 2D). sC5b-9, a marker of terminal pathway activation, increased significantly from Day 5 to Day 7 by 1.7-fold (*p* < 0.0001) (Figure 2E).

### 2.3. Soluble Platelet Activation Markers Increased during Storage

The soluble platelet activation markers thrombospondin-1 (TSP-1), transforming growth factor beta 1 (TGF-β1), neutrophil-activating peptide-2 (NAP-2), and platelet factor 4 (PF4) were analyzed in the T_0_ samples (n = 10), which were incubated for 15 min with the thrombin receptor activator peptide 6 (TRAP-6) for platelet activation. All four markers were significantly (*p* < 0.0001) elevated with TRAP-6 incubation on all days (Day 1–7) (Figure 3). The levels of TSP-1, NAP-2, and PF4 released by TRAP-6 incubation did not differ between Day 1 and any of the other days during the storage period. However, the released TGF-β1 level was significantly higher on Days 2–7 compared to Day 1 (*p* < 0.0001). All markers showed the lowest levels for the unstimulated platelet concentrates on Day 1. TGF-β1 increased significantly from Day 4 with a 1.9-fold change over the storage period (*p* < 0.001). NAP-2 increased significantly from Day 2 (fold change 2.5, *p* < 0.0001), and PF4 increased significantly from Day 5 (fold change 2.1, *p* < 0.0001). Although the level of TSP-1 increased 1.4-fold from Day 1 to Day 7, it did not reach the statistical significance threshold (*p* > 0.05).

### 2.4. Upregulation of Platelet Surface Markers

Platelet activation was further accessed by measuring the surface activation markers CD62P and CD63 in the T_0_ samples and in the samples incubated with phosphate-buffered saline (PBS), TRAP-6, and the TXA_2_R agonist U46619 during the storage period (Figure 4). The data were compared either to the unstimulated T_0_ control sample on each respective day (Figure 4A,B) or to the response on Day 1 for the respective treatment (Figure 4C–F). In comparison to T_0_, 15 min treatment with TRAP-6 and U46619, but not PBS, resulted in a significant increase in CD62P (Figure 4A) and CD63 (Figure 4B) (*p* < 0.0001–0.05). On Day 1, CD62P increased 28 times in response to TRAP-6 (*p* < 0.0001) and 33 times in response to U46619 incubation (*p* < 0.01). On Day 7, the corresponding increase was 13 times in response to TRAP-6 (*p* < 0.0001) and five times in response to U46619 incubation (*p* < 0.01). For CD63, there was a 5.3-fold increase in response to TRAP-6 (*p* < 0.0001) and 5.0-fold increase in response to U46619 incubation (*p* < 0.05). On Day 7, the increases were smaller at 3.8-fold for TRAP-6 (*p* < 0.0001) and 1.5-fold for CD63 (*p* < 0.05). When expressing the same data in relation to Day 1, we found that CD62P, but not CD63, was significantly (*p* < 0.01) increased over the storage period with a fold change of 1.7 on Day 7 (Figure 4C). The samples incubated with PBS had a 1.9-fold increase for CD62P (*p* < 0.001) and 1.1-fold increase for CD63 (*p* < 0.05) (Figure 4D). In contrast, compared to Day 1, the response to the activation stimuli of stored platelets significantly decreased; CD62P and CD63 significantly decreased by 0.81- (*p* < 0.001) and 0.76-fold (*p* < 0.05) in response to TRAP-6, respectively (Figure 4E) and 0.28- (*p* < 0.01) and 0.31-fold (*p* < 0.05) in response to U46619, respectively (Figure 4F).

### 2.5. Platelets Retained Their Adhesion and Fibrin Formation Abilities

The ability of platelets to adhere to collagen, as a measure of primary hemostasis, was assessed daily by flowing the platelets over a surface coated with collagen. Platelet adherence was recorded in real time, and the images taken between 165 and 195 s were used for analysis. There was no statistically significant loss in platelet binding throughout the storage period (Figure 5A,C). Fibrin deposition, analyzed between 390 and 450 s, showed similar results as those for platelet adhesion with no statistically significant change over the storage period (Figure 5B,D).

### 2.6. Marginal Changes in Inflammatory Cytokines and Growth Factors

A broad panel of inflammatory cytokines and growth factors was analyzed in the T_0_ samples (n = 10). Overall, the changes in cytokine levels were small. A total of 9 of the 27 analyzed markers, including IL-7, IL-9, TNF, eotaxin, IP-10, MIP-1β, FGF-2, PDGF, and RANTES, were found at detectable levels in the platelet concentrates (Figure 6A–I). Of these nine analytes, only IL-7 (Figure 6A) and TNF (Figure 6B) were significantly (*p* < 0.05) increased on Day 7 compared to Day 1; IP-10 (Figure 6C) was significantly increased on Day 4 (*p* < 0.05) and Day 7 (*p* < 0.01); and PDGF (Figure 6D) increased significantly from Day 4 and throughout the storage period (*p* < 0.05–*p* < 0.001). The most abundant increase was seen for PDGF, which increased 2.4-fold from 1091 ± 602 pg/mL (mean ± SD) on Day 1 to 2623 ± 1353 pg/mL on Day 7. The levels of IL-9, eotaxin, MIP-1β, FGF-2, and RANTES (Figure 6E–I) remained almost unchanged throughout the storage period.

## 3. Discussion

Complement activation with the accumulation of complement activation products in stored RDPs is associated with a risk for intra-bag platelet activation and cytolysis, including the release of damage-associated molecular patterns (DAMPs), which correlates with adverse transfusion reactions in patients [22]. Here, we comprehensively characterized the intra-bag complement activation and presence of inflammatory mediators, along with a functional assessment of the platelets in stored RDPs.

With the newly developed classical and lectin pathway-specific complement activation assays [27], we could, for the first time, show that complement activation in the RDPs is initiated via both the classical and lectin pathways, as demonstrated by the increased levels of the C1s/C1INH and MASP-1/C1INH complexes, respectively (Figure 2A,B). Both pathways are triggered via binding of pathway-specific recognition molecules: C1q in the classical pathway and MBL, ficolins, and collectins in the lectin pathway. Both the classical and lectin pathways depend on Ca^2+^ [23], and activation occurred in the RDPs despite the addition of citrate which can attenuate complement activation, indicating that it is not completely blocked [28]. Activation continued into the cleavage of C4 and C3 and the formation of the terminal complement sC5b-9 complex, also implying the release of C5a.

Since platelets are stored in plasma without refrigeration, complement activation is a factor to consider in platelet storage lesions, in accordance with previous studies [29,30,31,32]. De Wit et al. recently showed that storage under conditions where most of the plasma was exchanged with platelet additive solutions (PAS-E) significantly reduced the degree of complement activation [33]; however, they still observed complement activation, as we did in the present study. However, the potent inflammatory complement anaphylatoxins C3a and C5a have a short half-life before being inactivated to their des-Arg forms, and the C3b fragments lose much of their potency in solution. Thus, it is doubtful if the transfusion of platelets with complement activation products per se could mediate adverse transfusion reactions. At the ex vivo level, we found that platelet activation could potentiate the release of inflammatory cytokines, particularly IL-8, in human whole blood [34]; however, the clinical impacts of the activation of platelets used for transfusion, as observed in the current study, still need to be investigated.

Complement activation in stored platelet concentrates may stem from various underlying causes, each contributing to the overall activation process. First, any contact with an artificial surface lacking active complement regulation, such as the polymer surface of the storage bag, is prone to initiate complement activation. Zhong et al. showed that medical-grade PVC with two plasticizers (DEHP and DINCH) caused complement activation in human sera [35]. The alternative pathway amplification loop may substantially amplify any initial surface deposition of C3b if the surface is deficient in factor H or other alternative pathway regulators. Second, a small fraction of C3 in plasma continuously undergoes hydrolysis into C3(H_2_O) in the “tick-over” process [36]. C3(H_2_O) may form the soluble alternative pathway C3 convertase with factor B, i.e., C3(H_2_O)Bb, which activates C3. Elvington et al. showed that C3(H_2_O) is formed in sera stored at 22 °C [37], which makes this a plausible explanation for the increased complement activation in stored platelet concentrates. Third, activated platelets express several molecular patterns that are reported to trigger complement activation. The expression of gC1qR on activated platelets allows for the binding of the globular heads of C1q, triggering complement activation through the classical pathway [17]. P-selectin on activated platelets can attract C3b, leading to complement activation via the alternative pathway [18]. Additionally, C3(H_2_O) is adsorbed to activated platelets [20] and can interact with properdin for alternative pathway activation [21]. We found that platelet activation progressed over storage time, which makes the contribution of complement activation from activated platelets the most plausible explanation at later storage times. Fourth, the release of soluble DAMPs from activated or dying cells may trigger complement activation. CS-A, released from activated platelets and CpG in mitochondrial DNA, were reported to activate the complement cascade [16,38]. The fact that the concentrates are stored at 22 °C must be considered in relation to the above-mentioned studies, where most experimental data were generated at 37 °C. The proteolytic activation of complement factors depends on temperature, and activation at 20 °C is reduced but still present [39].

Of the 27 analyzed common cytokines in the Bio-plex assay, 9 were found at detectable levels. Five of these, IL-7, TNF, FGF-2, PDGF, and RANTES, were previously reported to be released from platelets or were found in platelet proteomic studies [40], whereas IP-10, IL-9, eotaxin, and MIP-1β were not, and may originate from leukocyte activation during RDP processing. This is also supported by the observation that only IP-10 was significantly increased during storage.

IL-7, TNF, and FGF-2 were, relative to PDGF and RANES, found in low abundance and might, thus, be of lower biological impact in this study. However, the platelet-dependent release of IL-7, a cytokine significantly elevated on Day 7, has been demonstrated to potentiate the release of IL-8 and MCP-1 from peripheral blood mononuclear cells [41]. PDGF and RANTES were found in substantial amounts and are, together with the other platelet markers TGF-β1, PF4, NAP-2, and TSP-1, indicators of intra-bag platelet activation. PDGF, TGF-β1, PF4, and NAP-2 showed increased levels as a function of storage time. PDGF has, like FGF-2, angiogenic effects and can stimulate endothelial cell proliferation [42]. RANTES, PF4, and NAP-2 are chemokines with multiple but predominantly pro-inflammatory functions. RANTES can recruit and activate several cell types, including monocytes and lymphocytes. RANTES was shown to bind inflamed endothelia and cause monocyte arrest [43]. The positively charged PF4 chemokine can interact with negatively charged structures, including heparan sulfate proteoglycans on the vessel wall, and modulate their hemostatic function [44]. NAP-2 is a chemokine that, among other functions, can attract and activate neutrophils and stimulate angiogenesis [45].

The level of spontaneous activation in the bag was significantly lower than the activation induced by TRAP-6, indicating that platelets throughout storage retained their ability to exert the platelet release reaction in response to PAR-1 stimulation. The release of TSP-1, PF4, and NAP-2 in response to TRAP-6 was stable throughout the seven days of storage. However, the TGF-β1 release in response to TRAP-6 was, intriguingly, significantly lower on Day 1 compared to Days 2–7. TGF-β1 is a pleiotropic cytokine with the immune-modulatory capacity to influence both tissue fibrosis and inflammation [46], including chemotactic and immunosuppressant effects [47]. Altogether, the observations demonstrated a shift in favor of pro-angiogenic and pro-inflammatory functions in cytokine pattern released from the stored platelet concentrates. However, the mechanisms underlying this observation remain unclear and are a subject for further investigation.

A striking effect of storage was the platelets’ reduced ability to respond to stimulation via TXA_2_R and PAR-1. Neither the effect of adherence to collagen or stimulation of fibrin formation was affected. The TXA_2_–TXA_2_R axis in platelets is essential for proper platelet activation and aggregation [48,49]. Our data indicate that stored platelets retained the ability to adhere to collagen, but may exhibit a reduced ability to form a stable platelet aggregate, primarily due to attenuated TXA_2_R activation.

Our data align with the results from previous studies that showed that prolonged ex vivo platelet storage produces platelet-storage lesions, i.e., attenuated platelet function and accumulation of inflammatory mediators, including complement activation products and cytokines, and platelet-derived DAMPs, all of which are transfused along with the platelets [7,22]. Allergic and febrile nonhemolytic transfusion reactions are adverse reactions following platelet transfusion. These reactions arise within hours after transfusion and are rarely life-threatening but cause discomfort through fever, chills, flushing, and/or vomiting and are associated with a prolonged hospital stay [50]. Allergic and febrile nonhemolytic transfusion reactions are linked to complement activation, platelet activation, and the release of platelet-derived DAMPs in platelet concentrates. DAMPs like HMGB-1 and mitochondrial DNA have been found to be elevated in platelet concentrates [51,52]. Upon transfusion, HMGB-1 can activate Toll-like receptor (TLR) 4 [53] on immune cells, and mitochondrial DNA can trigger innate immune activation, including complement activation [38,54] or TLR 9 activation [55]. Mitochondrial DNA may also promote complement activation in the bag during storage. Also, platelet-stored mediators, including soluble CD40L [56] and RANTES [57], have been linked to adverse transfusion reactions.

## 4. Materials and Methods

### 4.1. Preparation of Random Donor Platelet Concentrates

Human whole blood was collected from donors during routine donations at Kalmar Region Hospital, Sweden, and processed into units of erythrocytes, plasma, and interim platelet units (IPUs) using the Reveos^®^ system (Terumo BCT, Larne, UK). IPUs from four to five donors were pooled with PAS-E (Macopharma, Mouvaux, France) using the platelet pooling set with a leukocyte filter (Terumo BCT). Three concentrates were prepared from whole blood and the platelets were processed into IPUs on the same day; three were stored overnight before processing, and four were a mix between IPUs processed from overnight-stored and fresh whole blood. The concentrate was analyzed for bacterial contamination using the BACT/ALERT^®^VIROTU^®^ blood culture detection system (BioMérieux SA RCS, Lyon, France). Platelet count was assessed using an automated hematology system (Sysmex, Kobe, Japan) and leukocyte determination was conducted using flow cytometry (BD FACSCanto II, Franklin Lakes, NJ, USA). The RDPs were thereafter irradiated with 25 Gy gamma radiation (TrueBeam, Varian, Palo Alto, CA, USA). All commercial platforms mentioned above were used according to the manufacturer’s protocols.

### 4.2. Storage and Sampling

The platelet concentrates were stored in a 22 °C incubator (Forma platelet incubator, Thermo Electron Corporation, Waltham, MA, USA) with an agitator (Meloc Engineering Corp. Glendale, CA, USA). Every day for seven consecutive days, a sampling bag (Macopharma) was sterilely welded (TSCD, Terumo Corporation, Shibuya, Tokyo, Japan) to the 300 mL RDP, and a sample (≤8 mL) was isolated. The platelet concentrate and the sampling bag were separated using MACOSEAL PS2 (Macopharma), and the content of the sampling bag was transferred into a 15 mL polypropylene tube (Sarstedt, Nümbrecht, Germany). Platelet counts were performed using a Swelab Alfa (Boule Diagnostics, Spånga, Sweden).

### 4.3. Platelet Activation and Control

Each of the isolated platelet samples were divided into five separate aliquots. One aliquot was kept for the hemostatic analysis. The other four were recalcified to a final concentration of 23 mM CaCl_2_ to counteract the citrate in the PAS-E solution. Simultaneously, the thrombin inhibitor lepirudin (Refludan^®^, Celgene, Uxbridge, UK) was added to a final concentration of 50 μg/mL to prevent coagulation. One of the four aliquots (non-activated T_0_ sample) was subjected to the immediate addition of 10× stop solution which contained CTAD (0.11 M buffered trisodium citrate solution, 15 M theophylline, 3.7 M adenosine, and 0.198 M dipyridamole; Becton, Dickinson, and Company (BD) Franklin lakes, NJ, USA) and EDTA at a final concentration of 10 mM. The other three aliquots were incubated in 1.8 mL NUNC cryotubes (Thermo Fisher Scientific, Waltham, MA, USA) for 15 min at 37 °C in a water bath with either (i) the PAR-1 agonist TRAP-6 (25 μg/mL, Tocris, Bristol, Great Britain), (ii) the TXA_2_R agonist U46619 (3 μM, Tocris, No: 7E/249404), or (iii) Dulbecco’s PBS (Merck, Darmstadt, Germany) as the buffer control. After incubation, the reaction was terminated using the CTAD/EDTA stop solution. One aliquot of each of the non-activated T_0_ samples and the TRAP-6-activated samples was centrifuged at 3000× *g* for 20 min at 4 °C. The supernatants were isolated into 1.0 mL NUNC cryotubes (Thermo Fisher Scientific), snap-frozen in liquid nitrogen, and placed at −80 °C until further analyses. All the other samples were used for the flow cytometric analysis.

### 4.4. Complement Activation Markers

The markers of complement activation in the non-activated T_0_ samples were analyzed using ELISAs. The C1s/C1-INH complex, MASP-1/C1-INH complex, and C4c were analyzed using ELISA kits from Hycult Biotechnology (Uden, The Netherlands) according to the protocols from the manufacturer. C3bc and sC5b-9 were measured by in-house ELISAs that were previously described in detail in [58]. Absorbance was measured using a Tecan plate reader Sunrise (Tecan Nordic, Stockholm, Sweden), and the data were analyzed using the Magellan software, version 7.1 (Tecan Nordic AB).

### 4.5. Soluble Platelet Activation Marker Analysis

Soluble platelet activation markers in the non-activated T_0_ samples and the TRAP-6-activated samples were measured using R&D DuoSet^®^ ELISAs for PF4, NAP-2, TSP-1, and TGF-β1. All assays were performed following the manufacturer’s instructions. Absorbance was measured using a Tecan plate reader Sunrise, and the data were analyzed using the Magellan software, version 7.1.

### 4.6. Flow Cytometry

Platelets in the non-activated T_0_ samples, and in the samples incubated with either the PAR-1 agonist TRAP-6, the TXA_2_R agonist U46619, or PBS were analyzed. The platelets were stained with anti-CD42a-FITC (BD, clone: Beb1), and for expression of the surface activation markers CD62P and CD63, they were stained with anti-CD62P-PE (BD, clone: AK-4) and anti-CD63-PerCP (BD, clone: H5C6). A 5 μL volume of each antibody was mixed with 10 μL of the platelet sample and incubated for 20 min at 4 °C in the dark. The samples were fixed for 20 min at 20 °C using 475 μL of 0.2% formalin (Histolab, Gothenburg, Sweden) in PBS with 1% bovine serum albumin (BSA) (Merck). Flow cytometry was performed using a BD Accuri^TM^ C6 (BD); 10,000 CD42a-positive events were collected. The data were analyzed using FlowJo, version 10.7.1 (BD).

### 4.7. Platelet Adhesion and Fibrin Deposition

Ibidi uncoated µ-Slide flow cells VI 0.1 (Ibidi, Gräfelfing, Germany, catalog no. 80661) were coated with 30 μL 0.1 mg/mL Horm^®^ collagen (Takeda Pharmaceutical Company Limited, Tokyo, Japan) by overnight incubation at 4 °C. One hour before the start of the experiment, the channels were blocked with 1% BSA in PBS and put on top of a 37 °C-tempered glass plate (Oko lab, H601-Nikon-Ts2R-Glass-Flat, OKO lab, Napoli, Italy) mounted in a Nikon Eclipse Ts2R inverted microscope (Nikon Corporation, Tokyo, Japan). An in-house-manufactured tempered box enclosed the slide to create a temperature-controlled (37 °C) environment. Before the experiment, the channels were flushed with PBS to eliminate air bubbles from the flow channels. The isolated platelet concentrate was diluted to 250 × 10^9^ platelets per L in sterile PBS and aspirated into a 5 mL syringe (Soft-Ject, Braun, Melsungen, Germany), which was mounted in a syringe pump (Harvard Apparatus PHD Ultra, Harvard Apparatus, Holliston, MA, USA) and connected to the slide via Tygon tubing (Saint Gobain, Courbevoie, France, ref: ADF00002-C) and connectors (Ibidi, ref: 10824, 10825, and 394601). A 1 mL syringe (Omnifix-F, Braun) with 150 mM of CaCl_2_ for recalcification of the platelet solution was connected via a three-way valve BD Connecta (BD) just before the µ-Slide flow cell. The platelet solution was flowed over the Horm^®^ collagen-coated surface in the flow cell at 50 μL/min. Brightfield images were obtained every 5 s for the first five minutes, every 10 s for minutes five to ten, and every one minute from minutes 10 to 30, using the NIKON NIS-Elements Basic Research software, version 5.21.03.

In separate experiments, FITC-labeled fibrinogen was added to the platelet concentrates (final concentration of 50 μg/mL). FITC (Merck) had been conjugated to fibrinogen (Merck) through a 1 h incubation in 0.1 M carbonate–bicarbonate buffer, pH 9.2. The labeled fibrinogen was dialyzed in PBS before the experiment. The platelet concentrate with FITC-fibrinogen was flowed over the Horm^®^ collagen-coated surface as for the platelet adhesion experiment. Fluorescent images were obtained using a FITC filter, with the same frequency as for the platelet adhesion experiment.

### 4.8. Cytokine Analysis

The levels of 27 different cytokines (IL-1β, IL-1Ra, IL-2, IL-4, IL-5, IL-6, IL-7, IL-8, IL-9, IL-10, IL-12, IL-13, IL-15, IL-17, FGF-2, eotaxin, G-CSF, GM-CSF, IFN-γ, IP-10, MCP-1, MIP-1α, MIP-1β, PDGF, RANTES, TNF, and VEGF) in the non-activated T_0_ samples were analyzed using the Bio-plex Human Cytokine 27-Plex multiplex assay (Bio-Rad Laboratories Inc., Hercules, CA, USA) according to the manufacturer’s instructions. The measurements were performed using the Bio-Plex MAGPIX Multiplex Reader (Bio-Rad), and the data were analyzed using Bio-plex Manager 6.1 (Bio-Rad).

### 4.9. Image Analysis

Images from the platelet adherence and fibrin deposition experiments were analyzed using CellProfiler (version 4.2.5 Broad Institute. Inc., Cambridge, MA, USA). Each of the 113 consecutive images were converted to grayscale, and an area of 1200 × 1200 pixels at the center of the flow cell was analyzed. Bright clusters of pixels with at least a diameter of 5 pixels were defined as platelets/fibrin deposits. Platelet adherence and fibrin deposition were analyzed as percent area covered and as total light intensity, expressed as intensity units (IU). For the platelet read-outs, the mean value of the seven images captured at 3 min ± 15 s from the start of the experiment was used for the analysis. For the fibrin deposition read-outs, the mean value of the eight images captured at 7 min ± 30 s from the start of the experiment was used for the analysis.

### 4.10. Statistical and Data Analyses

Data visualization and statistical analysis were performed using GraphPad Prism 9 for Mac (San Diego, CA, USA). The tests employed were repeated measures one-way ANOVA with Dunnett multiple-comparison post-tests to analyze the time-dependent effects. Two-way ANOVA was performed on datasets including both time and treatment effects. A *p*-value < 0.05 was considered statistically significant.

### 4.11. Ethics Statement

This study was designed and performed according to the ethical guidelines of the Declaration of Helsinki. Informed written consent was obtained from the blood donors. The study was approved by the ethical committee of the Norwegian Regional Health Authority (ethical permit REK#S-04114, 2010/934).

## 5. Conclusions

In conclusion, platelet storage is associated with complement activation via both the classical and lectin pathways, which progress into forming the terminal C5b-9 complement complex. Storage was associated with platelet activation and a reduced ability of platelets to respond, in particular, to TXA_2_R activation. Reducing complement activation in stored platelet concentrates by producing and storing platelets under conditions that limit intra-bag complement activation or by introducing complement inhibitors, platelet activation could potentially be reduced. Thus, higher-quality platelet concentrates could be obtained, both in terms of retaining platelet function and attenuating the release of inflammatory mediators and damage-associated molecular patterns. At this stage, our results do not prove a causal relationship between complement activation, higher activation, and reduced responsiveness, and subsequent poor recovery and survival after transfusion. Such questions must be addressed using additional experimental approaches. Future studies should also focus on the underlying causes of complement activation during storage. In addition, clinical studies should also be performed to ask if complement activation is more likely to cause negative side effects after transfusion. There is also little insight into the possible impact of complement activation on the hemostatic behavior of platelets after transfusion, warranting an increased focus on in vivo studies of platelets after long-term storage.

## Figures and Tables

**Figure 1 ijms-25-01091-f001:**
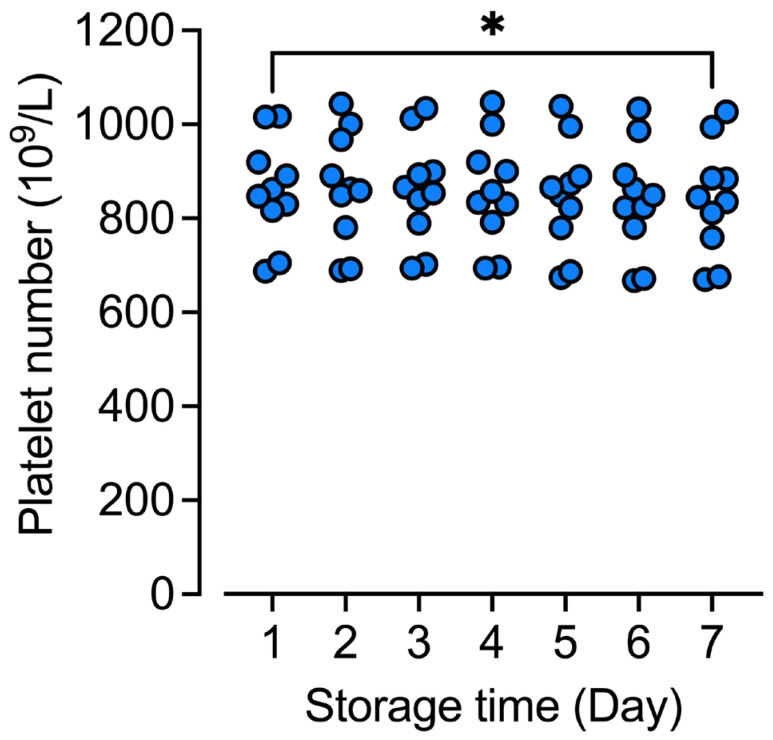
Platelet number in stored platelet concentrates. Platelet number was analyzed in EDTA-plasma aliquots from the stored platelet concentrates. Data are shown as platelet number per liter. Statistical differences were analyzed for all days from Day 2 to day 7 against Day 1 using repeated measures one-way ANOVA. * *p* < 0.05, n = 10.

**Figure 2 ijms-25-01091-f002:**
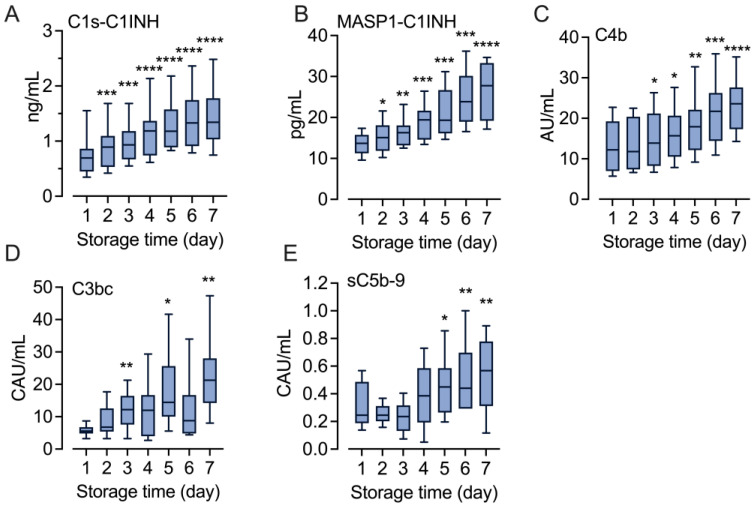
Complement activation in stored platelet concentrates. EDTA-plasma aliquots from the stored platelet concentrates were sampled daily and analyzed for a broad panel of complement activation markers: the classical pathway-specific C1s/C1-INH complex (**A**), the lectin pathway-specific MASP-1/C1-INH complex (**B**), the classical and lectin pathway marker C4c (**C**), the common marker C3bc (**D**), and the terminal pathway activation complex sC5b-9 (**E**). Data are shown in box and whisker plots with range showing 5–95 percentile, n = 10. The difference in the levels of activation markers between storage Days 2–7 and Day 1 were statistically tested using repeated measures one-way ANOVA. * *p* < 0.05, ** *p* < 0.01, *** *p* < 0.001, **** *p* < 0.0001.

**Figure 3 ijms-25-01091-f003:**
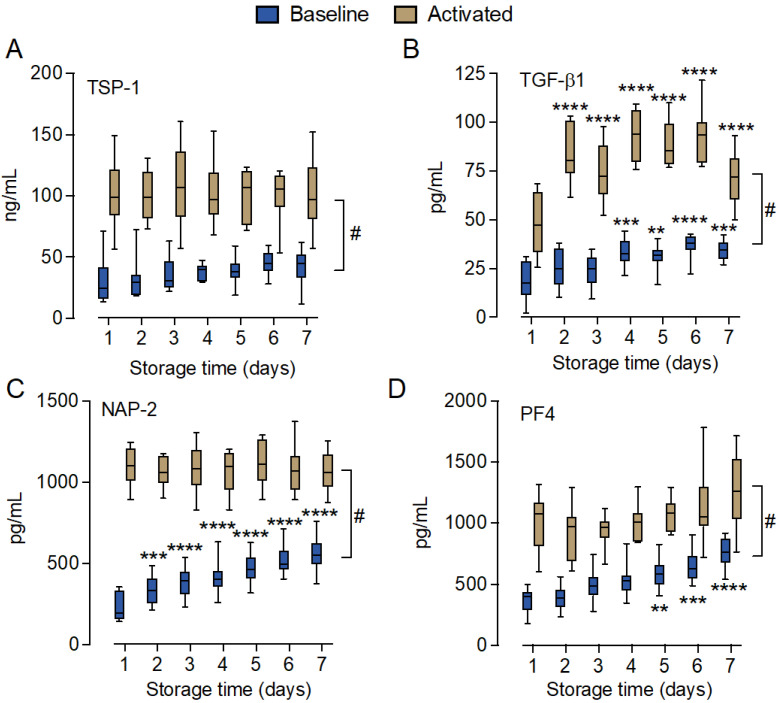
Soluble platelet activation markers in stored platelet concentrates. Levels of soluble platelet activation markers thrombospondin-1 (TSP-1) (**A**), transforming growth factor-beta 1 (TGF-β1) (**B**), neutrophil-activating peptide 2 (NAP-2) (**C**), and platelet factor 4 (PF4) (**D**) were measured in EDTA-plasma from non-activated platelet concentrate aliquots (blue bars) or platelet concentrate aliquots that were activated with TRAP-6 (brown bars). Aliquots were sampled daily from Day 1 to 7. Data are shown in box and whisker plots with range showing 5–95 percentile, n = 10. Statistical differences were analyzed using two-way ANOVA, either as a function of storage time (Days 2–7 in comparison to Day 1), ** *p* < 0.01, *** *p* < 0.001, **** *p* < 0.0001, or as a function of treatment. ^#^
*p* < 0.0001 for all days.

**Figure 4 ijms-25-01091-f004:**
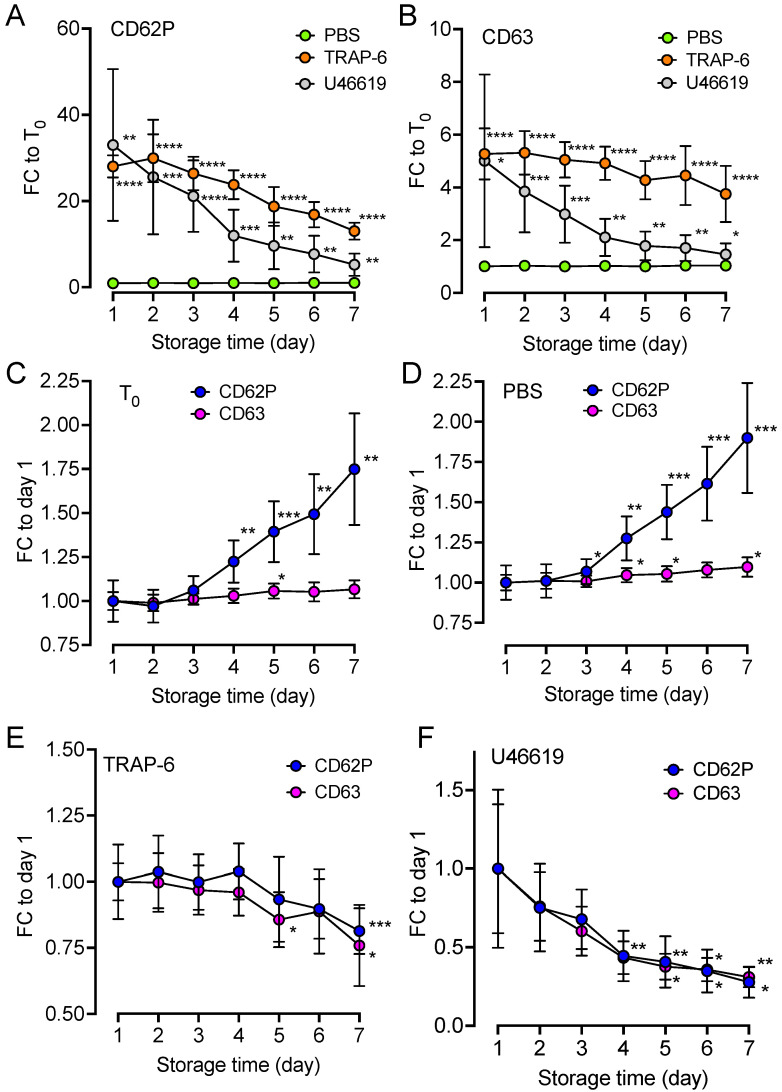
Platelet surface activation markers. Platelets were analyzed for surface activation markers CD62P and CD63 on all days of storage (Days 1–7) in response to PBS, TRAP-6 (a PAR-1 agonist), and U46619 (a TXA_2_R agonist). The data are shown as fold change (FC) either in relation to the unstimulated T_0_ control sample at the respective day for CD62P (**A**) and CD63 (**B**) or in relation to the response on Day 1 for each of the individual treatments: T_0_ (**C**), PBS (**D**), TRAP-6 (**E**), and U46619 (**F**). FC differences were statistically tested using a two-way ANOVA for the respective treatment in relation to T_0_ on each respective day (**A**,**B**) or respective treatment on Days 2–7 in comparison to the Day-1 response. Data are shown as mean ± standard deviation, n = 10. * *p* < 0.05, ** *p* < 0.01, *** *p* < 0.001, **** *p* < 0.0001.

**Figure 5 ijms-25-01091-f005:**
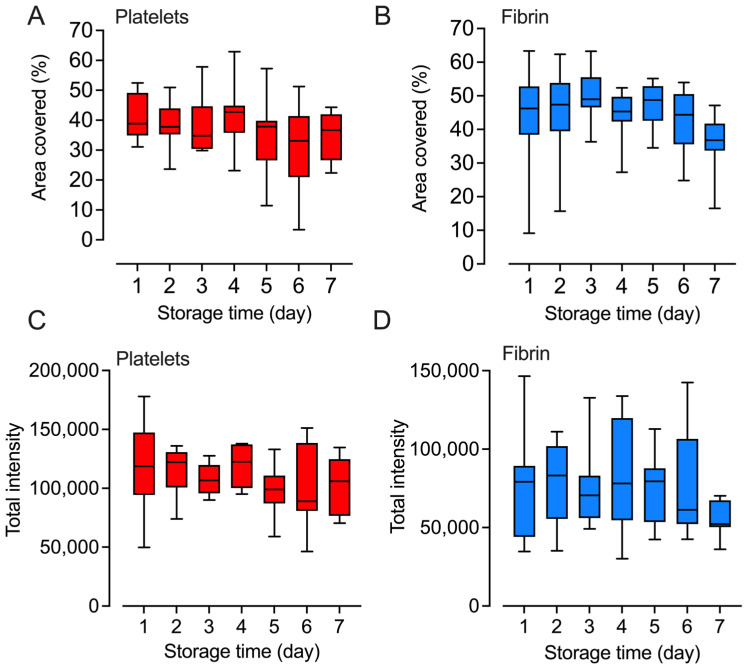
Hemostatic function. The stored platelet concentrates were analyzed for platelet hemostatic function in their ability to adhere to collagen (**A**,**C**) and stimulate fibrin formation (**B**,**D**) in a microfluidic system in real time, n = 9. The concentrates were analyzed daily (Days 1–7). Data for platelet adherence are shown for the first three minutes of analysis and fibrin deposition for the first seven minutes of analysis. Data are expressed as area coverage (**A**,**B**) or total intensity (**C**,**D**). Differences between storage Days 2–7 and Day 1 were statistically tested using repeated measures one-way ANOVA. There were no significant differences.

**Figure 6 ijms-25-01091-f006:**
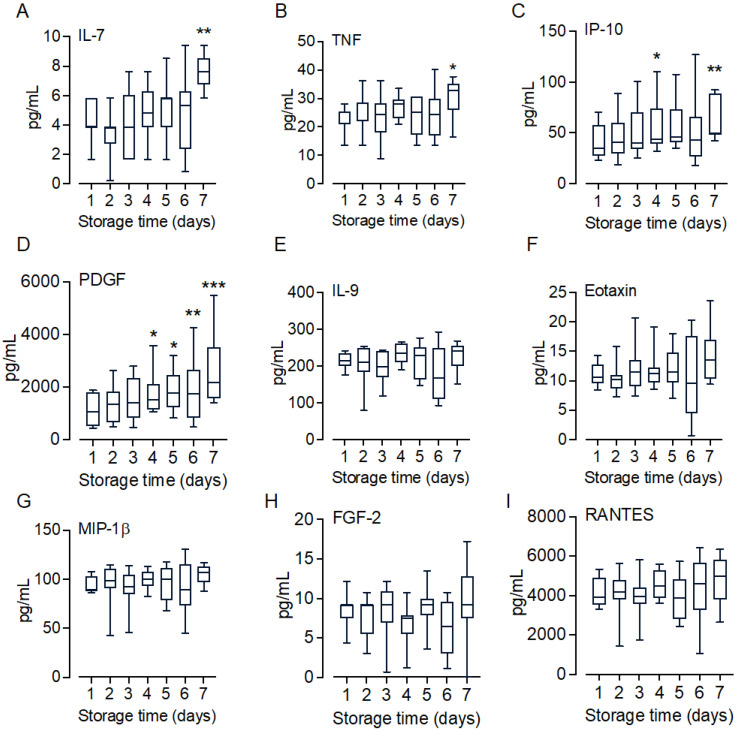
Cytokine levels in stored platelet concentrates. Of the 27 different cytokines analyzed, 9 analytes were detected in EDTA-plasma aliquots which were sampled daily from the platelet concentrates. Among these analytes, four cytokines, including IL-7 (**A**), TNF (**B**), IP-10 (**C**), and PDGF (**D**), showed significant increases during the storage period; the levels of five other cytokines, IL-9 (**E**), eotaxin (**F**), MIP-1β (**G**), FGF-1 (**H**), and RANTES (**I**), remained unchanged during this storage period. Data are shown in box and whisker plots with range showing 5–95 percentile, n = 10. The difference in cytokine levels between Days 2–7 and Day 1 were statistically tested using repeated measures one-way ANOVA. * *p* < 0.05, ** *p* < 0.01, *** *p* < 0.001.

## Data Availability

Data are contained within the article.

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
