# Peer review of "Storage of Transfusion Platelet Concentrates Is Associated with Complement Activation and Reduced Ability of Platelets to Respond to Protease-Activated Receptor-1 and Thromboxane A2 Receptor"

_ijms, 2024, doi:10.3390/ijms25021091_

Round 1
Reviewer 1 Report
Comments and Suggestions for Authors
The article entitled "Storage of transfusion platelet concentrates is associated with complement activation and reduced ability of platelets to respond to PAR-1 and TXA2R" is very interesting and provides important molecular and biochemical evidences of storage of transfusion platelet concentrates. I will write my comments in order of lines.
Line 77-78. The sentence is unclear; it is advisable to change it.
Line 388-404. In this section, the Authors describe the platelet proteome. In my opinion, the Authors should discuss in more depth the impact of cytokine pattern detected in the study.
Author Response
Reviewer 1:
Authors reply: We thank the reviewer for the encouraging review report:
- Thanks for noticing this incomplete sentence. This sentence is now re-written (line 77-81 in the revised manuscript).
- We agree with the reviewer’s suggestion and have now extended the discussion in relation also to the function of the cytokines, and impact of the cytokine pattern, with five new references (line 291-322 in the revised manuscript).
Reviewer 2 Report
Comments and Suggestions for Authors
Recommendation
Accept for publication after minor revision
Comments
The paper entitled: “Storage of transfusion platelet concentrates is associated with complement activation and reduced ability of platelets to respond to PAR-1 and TXA2R” (Manuscript ID: ijms-2805902) is an interesting paper evaluating the effects of storage time on complement activation and platelet function in routinely produced platelet concentrates. Ten platelet concentrates were stored at 22°C for seven days, with daily assessments for complement and platelet activation markers. Complement activation increased over the storage period, as indicated by elevated levels of various markers. Simultaneously, soluble platelet activation markers also increased in the concentrates. Platelet responsiveness to PAR-1 and TXA2R activation declined over the seven-day period, with a notable decrease in the ability to respond to TXA2R activation. Despite these changes, platelet binding to collagen remained largely unaffected during storage. The Authors conclude that complement activation during platelet storage correlated with increased platelet activation and reduced responsiveness, particularly to TXA2R activation. However, current results do not establish a causal relationship between complement activation, heightened platelet activation, reduced responsiveness, and subsequent poor recovery and survival after transfusion and additional experimental approaches are needed to address questions regarding the causal relationship.
The limitations include the need for additional experimental approaches to address questions regarding the causal relationship. In the discussion chapter, the underlying causes of complement activation during storage should be discussed in more detail.
Author Response
Reviewer 2:
Authors reply: We thank the reviewer for the comments and suggestions:
- We fully agree that no causal relationship is shown, and that any relationship must be proven by experimental data in future studies (as stated in lines 482-485 in the revised manuscript).
- We thank the reviewer for this comment and agree that the discussion lacked a section with underlying causes of complement activation during storage. We have added text accordingly, with five new references (lines 264-290).